# Photodynamic Therapy with Protoporphyrin IX Precursors Using Artificial Daylight Improves Skin Antisepsis for Orthopedic Surgeries

**DOI:** 10.3390/microorganisms13010204

**Published:** 2025-01-18

**Authors:** Tiziano A. Schweizer, Julia S. Würmli, Julia Prinz, Maximilian Wölfle, Roger Marti, Hendrik Koliwer-Brandl, Ashley M. Rooney, Vanni Benvenga, Adrian Egli, Laurence Imhof, Philipp P. Bosshard, Yvonne Achermann

**Affiliations:** 1Department of Dermatology, University Hospital Zurich, University of Zurich, 8091 Zurich, Switzerland; juliamichelleshannon@icloud.com (J.S.W.); yvonne.achermann@usz.ch (Y.A.); 2Department of Infectious Diseases and Hospital Epidemiology, University Hospital Zurich, University of Zurich, 8091 Zurich, Switzerland; 3Institute of Medical Microbiology, University of Zurich, 8006 Zurich, Switzerlandhkoliwer@imm.uzh.ch (H.K.-B.); vbenvenga@imm.uzh.ch (V.B.); aegli@imm.uzh.ch (A.E.); 4Analytica Medizinische Laboratorien AG, 8024 Zurich, Switzerland; 5Internal Medicine, Hospital Zollikerberg, 8125 Zollikerberg, Switzerland

**Keywords:** skin antisepsis, photodynamic therapy (PDT), daylight, 5-aminolevulinic acid, methyl-aminolevulinate, bacteria

## Abstract

Classical preoperative skin antisepsis is insufficient in completely eliminating bacterial skin colonization for arthroplasty. In contrast, photodynamic therapy (PDT) with red light and methyl-aminolevulinate (MAL), combined with skin antisepsis, led to the absence of bacterial growth in healthy participants, though with local skin erythema, posing an obstacle for orthopedic surgery. Therefore, we explored whether artificial daylight PDT (PDT-DL) was superior to red light. Twenty healthy participants were allocated to either 5-aminolevulinic acid-(5-ALA) PDT-DL (n = 10) or MAL-PDT-DL (n = 10) before antisepsis with povidone-iodine/alcohol. Skin swabs from the groin were taken to cultivate bacteria at baseline, after PDT-DL, and after the subsequent antisepsis. Additional swabs were taken on day 4 before and after antisepsis without PDT. The contralateral groin of each participant and of ten additional healthy volunteers served as the control (n = 30). In selected participants, 16S rRNA-based amplicon deep sequencing was performed. All participants showed a baseline bacterial colonization. After a PDT-DL with skin antisepsis, bacterial growth occurred in three (30%) and in one (10%) participants with 5-ALA and MAL, respectively, compared to the sixteen (55%) participants in the control group. On day 4, three (30%) participants per group showed positive cultures post antisepsis. Adverse effects were reported in six (60%) and zero (0%) participants for 5-ALA- and MAL-PDT-DL, respectively. The skin bacteriome changes correlated with the bacterial culture results. The MAL-PDT-DL with skin antisepsis significantly increased bacterial reduction on the skin without adverse effects. This offers an opportunity to prevent infections in arthroplasty patients and reduce antibiotic use, thus contributing to antibiotic stewardship goals emphasized in the One Health approach.

## 1. Introduction

The use of orthopedic implants has been steadily rising over the last few decades due to the growing demand for peri-prosthetic joints in the growing elderly and frail population [1,2]. Peri-prosthetic joint infections (PJIs) are one of the most feared complications after the implantation of an orthopedic device. Despite the increased emphasis and efforts for infection prevention, infections among hip and knee arthroplasties have been increasing [3]. PJIs occur in up to 1–2% of these operations [1] despite standard preoperative skin antisepsis and perioperative antibiotic prophylaxis, and are associated with morbidity and adverse effects due to the extended antibiotic prescription [4]. This prolonged exposure is among the factors contributing to global antibiotic resistance development. In addition, the cost per one episode of PJI treatment mount up to more than CHF 45,000 [1]. The most commonly isolated microorganisms in PJIs are staphylococci, streptococci, and aerotolerant anaerobic bacteria, such as *Cutibacterium acnes* or *Cutibacterium avidum* [5], all of which constitute part of the normal human skin flora. The majority of PJIs are acquired during surgery, with the assumption that the bacteria colonizing the skin surface contaminate the orthopedic implant during surgery and cause PJIs [6].

Skin antisepsis at the incision sites is the primary intervention to prevent commensal bacterial infection into deep tissues during surgery. However, recent studies have shown that the current standards for skin antisepsis are not effective enough to sterilize the skin completely before arthroplasty, with positive bacterial skin cultures obtainable in up to 70% of cases [7,8,9,10]. It has been suggested that bacterial colonization in deeper skin, such as within sebaceous or sweat glands, which are located 2–4 mm below the surface [11], contribute to the inefficiency of topical skin antisepsis [7]. The incomplete eradication of bacteria in these deeper skin layers might harbor the risk of causing infection during orthopedic surgery. Therefore, skin antisepsis approaches have to consider the deep skin bacterial reservoir. Different approaches to improve antisepsis have been examined throughout the years, such as benzoyl peroxide, though none have translated to clinical application [10,12,13]. One interesting potential approach to target these deep-seated bacteria is photodynamic therapy (PDT). The antimicrobial effect of PDT relies on photophysical and photochemical reactions [14]. Antimicrobial PDT (aPDT) is not a recent discovery, but it has not achieved the same success in the field of skin antisepsis as it has in other therapeutic areas, where it has become routine clinical practice, such as in oncology [15], dermatology [16] and dentistry [17,18,19]. However, aPDT has gained renewed interest thanks to the success obtained in these areas, as well as the urgent need to curb the development of antimicrobial resistance, by employing alternative antimicrobial strategies as outlined under the proposed antibiotic stewardship program of the One Health approach [20]. The principle of aPDT relies on the use of three components in combination, i.e., (1) a photosensitizer (PS), which is a non-toxic photoactive molecule, (2) light of the appropriate wavelength to activate the PS and (3) oxygen, transformed into a highly reactive oxygen species upon energy transfer from the light-activated PS [21]. These highly reactive oxygen species then cause damage to cellular structures, ultimately resulting in cell death [22].

The most well-studied application for dermatological purposes employs the PS prodrugs 5-aminolevuling acid (5-ALA) and its ester derivative methyl-aminolevulinate (MAL) [23,24,25,26,27]. Both 5-ALA and MAL have no photoactive properties, but, when applied exogenously, they can act as a precursor of photosensitive porphyrins, such as protoporphyrin IX (PpIX) [28]. The accumulation of PpIX is stronger in cells with a high metabolic turnover, such as cancer cells or bacteria [29]. Bacteria have been shown to efficiently metabolize 5-ALA or MAL into PpIX as well as other photoactive compounds [15,28,30], thereby allowing the effects of aPDT to occur upon light application. While PpIX can be excited by both red (600–740 nm) or green (450–580 nm) light [31], red light allows for a deeper penetration of up to 5 mm into tissues [11], the area where sebaceous and sweat glands can be found. Of note, it has been shown that Gram-positive bacteria, the bacteria most commonly causing PJS, are more susceptible to aPDT, as compared to Gram-negative bacteria [28].

We recently showed that a PDT with MAL and red light (633 nm, 13 min, 40 J/cm^2^) improves skin antisepsis [32]. Immediately after MAL-PDT combined with skin antisepsis treatment, no bacterial growth was detected in all of the tested participants. Fifty percent of the participants in the control group receiving only skin antisepsis still showed bacterial growth. However, MAL-PDT with red light led to transient skin erythema, which is an obstacle for immediate surgery for the implantation of an arthroplasty.

In this follow-up study, we aimed to compare the application of 5-ALA to MAL with artificial daylight PDT (PDT-DL, 400–700 nm, 30 min, 10 J/cm^2^) in terms of both bacterial growth reduction and local adverse skin effects in a clinical trial of healthy volunteers.

## 2. Materials and Methods

### 2.1. Participants and Study Design

This clinical trial was conducted as an open-label, single-center study. From January 2021 to July 2023, 30 healthy male and female participants (≥18 years) were recruited for PDT treatment in combination with skin antisepsis or skin antisepsis treatment only. The exclusion criteria were age <18 years, an existing pregnancy or lactation period, predisposing conditions leading to an increased susceptibility to photo dermatosis, antibiotic treatment within 14 days of the first study visit, treatment with retinoids within the last 6 months and an inability to follow the instructions of the study leaders [32]. Further, participants with a history of photosensitive disorders or with Fitzpatrick skin phototype V and VI were also excluded due to the potentially higher risk of developing inflammatory reactions to light exposure, as well as the potentially thicker skin layers, potentially limiting the effect of PDT to reach deep-seated skin bacteria. The study was approved by the local ethical committee (BASEC No. 2020-02183 and 2022-01272). Study participants signed an informed consent, and we adhered to the Declaration of Helsinki for ethical principles for medical research involving human subjects [33]. The study was approved by the local ethical committee and submitted to Clinicaltrials.gov, with the clinical trial number NCT04618276 and NCT05676801.

Baseline participant characteristics, such as age, BMI and comorbidities, were recorded on the first day of the clinical visit. The study design is shown in Figure 1. A total of 30 participants were enrolled, whereof 10 participants were allocated to either the 5-ALA- or the MAL-PDT-DL arm and 10 participants were allocated to the control group receiving skin antisepsis only. In the 5-ALA- or MAL-PDT-DL arm, the contralateral leg of each participant also served as a control with antisepsis only, resulting in a control group of 30 participants. The study participants underwent follow-up examinations to assess the adverse effects and were asked about the occurrence of redness (grade 1–3), pain (measured on a visual analog scale of 0–10), skin irritation, itching or any other abnormality. They were also asked about the need for medications to alleviate any reported symptoms.

### 2.2. Skin Swabs

Skin swabs were taken as described previously [32]. For the PDT treatment that was performed in the right groin, skin swabs were taken immediately before PDT, after PDT and after standard skin antisepsis treatment on day 1, and additionally on day 4, before and after antisepsis without PDT to check for bacterial growth on human skin. Control swabs were taken from the left groin area before and after antisepsis alone on day 1 and day 4. The swabs were taken below the anterior superior iliac spine using a scraping technique with sterile blades. Loosened epidermal scales on the blade and skin were transferred to an eSwab^TM^ Collection and Transport System with liquid Amies medium (Copan Group, Brescia, Italy).

### 2.3. Photodynamic Therapy with Artificial Daylight (PDT-DL)

PDT-DL treatment was performed at the Department of Dermatology of the University Hospital of Zurich, Switzerland. For the 5-ALA-PDT study arm, the participants received two transdermal patches containing 8 mg of 5-ALA each (Alacare, Gebro Pharma AG, Liestal, Switzerland) and were asked to apply the patches next to each other 4 h before the PDT session in the right groin area. For the MAL-PDT study arm, a topical MAL (Metvix crème 160 mg/g, 2 g tube, Galderma SA, Lausanne, Switzerland) was applied to a 5 cm × 10 cm area in the right groin area in a sheer layer, covered by a light-impermeable bandage and incubated for 1 h. The 5-ALA patches or topical MAL cream were then removed and the skin area was exposed to indoor polychromatic artificial daylight using Medisun PDT 1200 (Schulze & Böhm GmbH, Brühl, Germany) for 30 min with an energy dose of 10 J/cm^2^. The Medisun PDT 1200 employs a continuous light spectrum between 400 and 700 nm with the simultaneous application of all of the colors of visible light.

### 2.4. Skin Antisepsis

The skin antisepsis treatment of the groin was performed three times according to the standard operation room procedure using Betaseptic (3.24% povidone-iodine, 38.9% 2-propanol and 38.9% ethanol 96%, Mundipharma, Cambridge, UK) with a one-minute interval between applications.

### 2.5. Microbiological Analysis

The skin swabs were evaluated for bacterial growth from the eSwabs on Columbia sheep blood agar plates without antibiotics (bioMérieux, Mary-l’Etoile, France) and colistin-nalidixic acid blood agar (bioMérieux, Mary-l’Etoile, France) plates for aerobic cultivation. Brucella agar plates (in-house sheep blood agar plates with hemin added Vitamin K1 provided by the Institute of Clinical Microbiology, Zurich, Switzerland) were used for anaerobic cultivation using GENbags (bioMérieux, Mary-l’Etoile, France). Agar plates were cultured for 7 days at 37 °C. Each bacterial colony was identified using matrix assisted laser desorption/ionization-time of flight (MALDI-ToF) mass spectrometry using the MALDI Biotyper^®^ smart(Bruker Daltonics, Bremen, Germany) using a 337 nm wavelength nitrogen laser. MALDI-ToF MS is well explained in the literature [34,35,36]. This technique is considered as the gold standard in clinical microbiology for rapid microbial identification [37]. In short, we spotted all strains onto the MALDI target plate and overlaid them with 1 μL of 70% formic acid. After drying at room temperature, we covered and dried the spots with 1 μL α-cyano-4-hydroxycinnamic acid (HCCA) matrix according to the manufacturer’s instructions (Bruker Daltonics, Bremen, Germany). Each spot was read in the positive ion mode (detecting cations) with a m/z detection range of 2000 to 20,000 Da, which is the range that captures the most ribosomal proteins needed for bacterial identification. The species of each mass spectrum was identified using the MALDI Biotyper^®^ software package (software version: MBT Compass 4.1.100.10) with the MBT reference library (version BDAL-12.0, including 11,897 species) and default parameter settings. The scoring criteria for species-level identification is Log(score) ≥ 2.0 and for genus-level identification, Log(score) ≥ 1.7 and < 2.0.

Bacterial growth was reported either semi-quantitatively (“sporadic” = 10^3^–10^4^ CFUs, “moderate” = 10^4^–10^5^ CFUs, ”abundant” = 10^5^–10^6^ CFUs) or as numbers of colonies and the corresponding quadrant of the streak [32]. All processes are part of the daily clinical routine of an ISO/IEC 17025 accredited diagnostic laboratory [38].

### 2.6. Metagenomic Analysis

The swabs were subjected to DNA extraction using the Maxwell RSC Buccal Swab DNA kit (ProMega, Madison, WI, USA). The DNA was quantified using the Qubit Fluorometer (Thermo Fisher Scientific, Waltham, MA, USA). The negative controls (DNA extraction and PCR negative control) and positive control (ZymoBIOMICS Microbial Community Standard (Zymo Research, Irvine, CA, USA) were included in the sequencing run. DNA libraries were generated using the QiaSeq 16S/ITS panels (Qiagen, Hilden, Germany) which employs phased primers targeting the V1–V9 regions of the 16S rRNA gene and the ITS region for fungal characterization. Primers are available from QIAGEN upon request. The PCR conditions used are specified in the QIAseq 16S/ITS Screening Panel protocol. All PCRs were performed on a Biometra 48 TRIO thermocycler (Analytik Jena AG, Jena, Germany). In short, the PCR-applied protocol consists of a 3-step cycling with 30 s denaturation at 95 °C, 30 s annealing at 50 °C and 2 min extension at 72 °C, which was repeated for 35 cycles due to a general low abundancy of DNA in skin samples. A final extension step takes place for 7 min at 72 °C before the holding step in PCR is performed at 4 °C. A positive amplification control was included for the PCR but was not subjected to sequencing. PCR amplicons were subjected to a quality assessment using the Fragment Analyzer 3200 (Agilent Technologies, Santa Clara, CA, USA) with the high sensitivity 1–6 kbp next-generation sequencing fragment kit (cat.# DNF-474-0500). Single libraries were quantified with the Qubit Fluorometer (Thermo Fisher Scientific, Waltham, MA, USA), diluted to 2 nmol and pooled in equimolar concentration. The diluted pooled library (10 pmol) was loaded on the MiSeq (Illumina, San Diego, CA, USA) for paired-end 301 base pair sequencing using V3 chemistry (600 cycles) (Illumina, lot# 20795255/20769174). The V3–V4 region of the 16S rRNA gene was analyzed in QIIME2 version 2024.2 [39]. The demultiplexed paired-end sequences (average sequencing depth of 22067) were subject to DADA2 [40] for quality filtering, denoising, and chimera removal. Briefly, a naïve Bayes classifier was trained on the SILVA 138 (released on 16 December 2019) [41] full-length reference sequences and then used to taxonomically annotate the sequences in the current study. Amplicon sequence variants (ASVs) classified as chloroplasts were removed as these are likely false positives due to their presence in abundances of approximately 0.1% in two patient samples. Low frequency or rare ASVs (those that contribute less than 0.1% of the average sequencing depth) were also removed. Mitochondrial reads or other eukaryotic reads were not present in the dataset. To eliminate sequencing depth bias, the samples were rarefied or evenly sampled to 3031 sequences per sample, which was the lowest number of sequences that retained all the samples. Diversity analyses were conducted at the genus-level. The genus-level taxonomy is plotted as histograms.

### 2.7. Statistics

Data visualization and quantitative analyses were performed using Prism 9.2.0 (GraphPad Software, San Diego, CA, USA) and QIME2 version 2024.2 [39]. The data were assessed for normality (Gaussian distribution) based on a conclusion drawn from results obtained from a Shapiro–Wilk, Anderson–Darling, D’Agostino–Pearson and Kolmogorov–Smirnov normality test. Fisher’s exact test was used to determine the statistical significance between the treatment outcomes.

## 3. Results

### 3.1. Patient Demographics

A total of 19 out of 30 (63%) healthy participants in this study were female. The median age of the participants was 30 years (range: 19–93 years) and the median BMI was 24.5 kg/m^2^ (range: 17.6–33.5 kg/m^2^).

### 3.2. Baseline Bacterial Skin Colonization of the Participants

All 30 participants were colonized with different bacteria at baseline sampling, including coagulase-negative staphylococci (CoNS: *Staphylococcus caprae/capitis*, *Staphylococcus epidermidis*, *Staphylococcus hominis*, *Staphylococcus warneri/pasteuri*, *Staphylococcus hominis*), *C. acnes*, *Micrococcus luteus*, *Bacillus* spp. and others (Figure 2A).

### 3.3. Standard of Care and PDT-DL Antisepsis Effect on Bacterial Skin Colonization

Directly after the 5-ALA-PDT-DL, bacterial growth on the skin, as determined per microbiological culture and the subsequent MALDI-TOF-MS analysis, was found in eight (80%) participants. After an additional skin antisepsis treatment, bacterial growth was obtained in three (30%) participants (Figure 3), with colonizing *C. acnes* (n = 2) and *Brachybacterium* sp. (n = 1). Directly after MAL-PDT-DL treatment, bacterial growth was obtained in four (40%) participants, while after an additional skin antisepsis treatment, the bacterial growth of *S. epidermidis* was obtained in only one (10%) participant. In the control group, after antisepsis treatment alone, skin colonizing bacteria were found in 16 (55%) participants. Overall, 5-ALA-PDT-DL and MAL-PDT-DL combined with antisepsis led to a 46% and 82% reduction in participants with a positive bacterial culture, compared to standard antisepsis alone. After standard skin antisepsis three days post-treatment, both study groups showed a non-significant 14% reduction in positive bacterial cultures as compared to the control group (Appendix A).

The bacterial species remaining after PDT-DL and/or antisepsis were heterogeneously distributed, but predominantly comprised *C. acnes*, *S. epidermidis*, *S. hominis*, and *M. luteus* (Figure 2 and Appendix A).

### 3.4. Adverse Effects of PDT-DL

PDT-DL was, in general, well tolerated. In the 5-ALA-DPT-DL arm, none of the participants reported pain, while four (40%) participants reported slight erythema after PDT. Itching was reported by one (10%) participant and both itching and erythema during PDT were reported by two (20%) participants. In the MAL-DPT-DL arm, none of the participants reported pain, erythema, or itching at any time during or after PDT (Table 1).

### 3.5. MAL-PDT-DL Alterations of the Skin Bacteriome

To gain an in-depth understanding of MAL-PDT-DL on the skin bacteriome, we performed metagenomics analyses from three participants at four timepoints: the day 1 baseline, after MAL-PDT-DL, after additional antisepsis and the day 4 baseline. A Bray–Curtis dissimilarity analysis revealed three distinct clusters (unpublished material): day 1 baseline samples and participant 11’s day 4 baseline sample clustered away from the post-treatment samples, suggesting differences in bacteriome composition between the baseline and treatment samples. The treatment samples (MAL-PDT-DL with and without additional antisepsis) appeared to also cluster away from the day 4 baseline samples, suggesting another change in bacterial composition between these time points. Regarding the bacteriome composition, at the day 1 baseline, the Firmicutes are highly abundant, including *Staphyloccoccus* spp. and *Enterococcus* spp., which correlates with the bacterial culture of these samples (Figure 4A). Importantly, while *Cutibacterium* spp. was detected, it was at an extremely low abundance and therefore not represented. Also, the cultures of these samples were negative for the growth of *Cutibacterium* spp. The post-treatment samples exhibit a loss of Firmicutes with a high median richness and diversity compared to the baseline (Figure 4B,C). Participant 11’s day 4 baseline sample returned to a baseline-like state with high abundances of Firmicutes, mainly *Staphylococcus* sp., with low richness and diversity.

## 4. Discussion

Multiple studies have shown that current skin antisepsis procedures are inadequate for the complete elimination of bacterial growth on the skin. On average, ≥50% of patients show positive skin cultures after antisepsis [9,10,32]. In line with these results, 55% of the participants in our study were still positive for bacterial growth on the skin after the same skin antisepsis procedure. We demonstrated that PDT-DL, when combined with either 5-ALA or MAL alongside standard skin antisepsis procedures, leads to a significant reduction in positive bacterial skin cultures, compared to standard antisepsis alone. However, this effect was not long-lasting. Three days after PDT (day 4), the bacterial growth on the skin had returned to baseline levels, similar to that on day 1, in both groups.

The photosensitizer, MAL, showed a more pronounced immediate effect than 5-ALA, with bacterial growth remaining in only 10% versus 30% of participants. It was previously shown that MAL offers superior selectivity and penetration into skin thick basal cell carcinoma, as well as sebaceous glands, as compared to 5-ALA [42,43]. Therefore, it is possible that similar effects took place in our study, such as the increased selectivity and accumulation of MAL over 5-ALA in bacteria or sebaceous glands, resulting in the enhanced effect of MAL- over 5-ALA-PDT. However, it is also important to note that one in vitro study showed that 5-ALA- or MAL-PDT at the same concentration, although with a red light source, led to a similar reduction in planktonic *E. faecium* [44]. Therefore, another reason for the observed difference in our study could be due to the varying photosensitizer dose or their mode of application. While 5-ALA at a dose of 16 mg was applied as a patch by the participants themselves for a period of 4 h, MAL at a dose of approximately 100 mg was applied as a cream by the study physician for a period of 1 h. While certain bacteria, such as *S. epidermidis* and *C. acnes*, produce their own endogenous photoactive porphyrins [45] and could therefore be targeted without the application of any prodrug PS, other bacteria involved in PJIs are not known to produce photoactive PS. In addition, while it is known that the amount of PS produced by *C. acnes* is sufficient to be used with aPDT for acne treatment [46], it is not known whether the amount of PS produced by bacteria in the deep skin in the groin is sufficient to elicit phototoxic reactions. Therefore, the application of a prodrug PS, such as MAL or 5-ALA, is advised to be used for the application of aPDT for skin antisepsis.

Despite a very good reduction in growing bacteria on the skin in this study, we were unable to achieve the complete elimination of bacterial growth in all participants, which had previously been demonstrated in our previous clinical trial by Waldmann et al. using MAL-PDT with a red light source [32] instead of artificial day light. Red light is known to penetrate deeper into tissues [47] and, therefore, might lead to an enhanced reduction in bacteria found in the dermal layers of the skin [7,10]. However, MAL-PDT with red light is also associated with local cutaneous adverse effects after PDT, which we previously published [32], challenging its immediate use prior to arthroplasty procedures.

In our current MAL-PDT-DL arm, no adverse effects were observed, whereas minor local adverse effects occurred in the 5-ALA-PDT-DL arm. These results align with a previous study, which reported less pain in patients with actinic keratosis treated with MAL-PDT compared to 5-ALA-PDT with red light [48]. We hypothesize that the higher number of side effects could be due to the length of application: the 5-ALA patches were applied to the skin for 4 h, whereas the MAL cream was only applied for 1 h. The extended application of 5-ALA may have contributed to its broader uptake, including by eukaryotic cells, potentially triggering adverse effects.

In this study, we not only looked at the absolute reduction in the number of bacteria, but also looked, in detail, at bacterial composition on a cultural and genetic level. Using conventional culture techniques, we found skin bacteria representing those most commonly involved in chronic PJIs, such as CoNS (predominantly *S. epidermidis*) and *C. acnes* [1] after PDT and skin antisepsis. These bacteria might reside in deeper skin reservoirs [7], as assessed in other studies performing skin antisepsis procedures [7,8,10]. We found the same bacteria using artificial daylight compared to our previous study using red light PDT [32].

We performed a metagenomics analysis to determine the influence of MAL-PDT-DL on the skin bacteriome composition. We observed similar results looking at the culture results compared to the skin bacteriome composition using a metagenomics analysis but we gained more detailed information: while the baseline samples demonstrated low bacterial diversity with mainly *Staphylococcus* spp. as the dominant organism, MAL-PDT-DL and antisepsis disrupted the skin bacteriome with a potentially low bacterial density, low Firmicutes relative abundances, high microbial richness and Shannon diversity. The day 4 baseline samples appeared to differ in microbial diversity relative to the treatment samples, but did not return to an exact day 1 baseline state, except for one participant, suggesting potential long lasting skin bacteriome alterations. These skin bacteriome alterations will require further investigation in larger studies, since it was shown that the depletion of specific bacterial species could lead to the over-colonization by other species. Specifically, it was shown that the depletion of *S. epidermidis* could lead to the over colonization by *C. acnes* [49]. Therefore, it remains to be investigated whether MAL-PDT-DL-induced alterations will not lead to shifts in the bacteriome towards over-colonization with potentially more difficult-to-treat bacteria. It should be noted that we only included healthy subjects without skin diseases in the study. The bacteriome composition of patients with acne can differ from that of healthy participants, as shown in a study using 5-ALA-PDT in severe acne patients [48].

We did not observe a long-lasting effect of PDT-DL, as also shown in the study by Waldmann et al. [32]. The bacterial culture from day 4 before antisepsis had returned to baseline levels in both groups. It seems that the bacteriome quickly returns to the previously disinfected skin. It was previously shown that the bacterial repopulation of the shoulder skin occurred in 50% of participants at 30 min post chlorhexidine treatment and all participants had positive bacterial skin cultures at 240 min post treatment [50]. However, in our study, it remains unknown how quickly the recolonization after PDT occurs, since we only repeated bacterial cultures three days post-PDT. These findings highlight the rapid recolonization of sterilized areas either from deeper or surrounding skin. Based on our findings, PDT-DL application for enhanced skin sterility can only be used immediately prior (i.e., ideally within an hour) to arthroplasty in order to obtain its benefit. Since this might cause certain caveats within a narrow and timely surgical plan, further studies might explore whether multiple PDT-DL applications over various days could further modulate the skin bacteriome and lead to less rapid recolonization.

Despite these interesting findings, our study entails some limitations. First, we only assessed the effects of PDT-DL on healthy participants. Patients undergoing arthroplasty potentially present with a different skin bacteriome compositions due to multimodal factors, such as age, comorbidities and BMI [10]. Second, we did not perform any deep skin sampling to investigate the bacterial colonization of deep skin. The scalpel scrapping technique allows for the retrieving of at least some bacteria from deeper skin [32]. Third, the bacteriome analysis has some limitations: despite the correlation of a high relative abundance of the identified genus-level taxa and the bacteria recovered in culture, the genus-level taxa identified in the post-treatment samples must be interpreted with caution. Due to the low bacterial density in these samples and the requirement of increased amplification cycles, contaminations potentially arising from the hospital environment might influence these results [36,51]. In addition, errors might arise from the possible incompatibility of the current taxonomy and the information in the metagenomics databases used [52].

In conclusion, we demonstrated that preoperative PDT-DL in combination with normal skin antisepsis is a promising novel approach that may help prevent PJIs by reducing the growth of skin bacteria. This harbors the chance to drastically reduce antibiotic prescription and thereby contribute to the global efforts of antibiotic stewardship under the One Health approach. MAL-PDT-DL showed a promising result regarding efficacy and safety in comparison with 5-ALA-PDT-DL or MAL-PDT and red light. This study lays the foundation for a MAL-PDT-DL prospective clinical trial with patients undergoing orthopedic surgery.

## Figures and Tables

**Figure 1 microorganisms-13-00204-f001:**
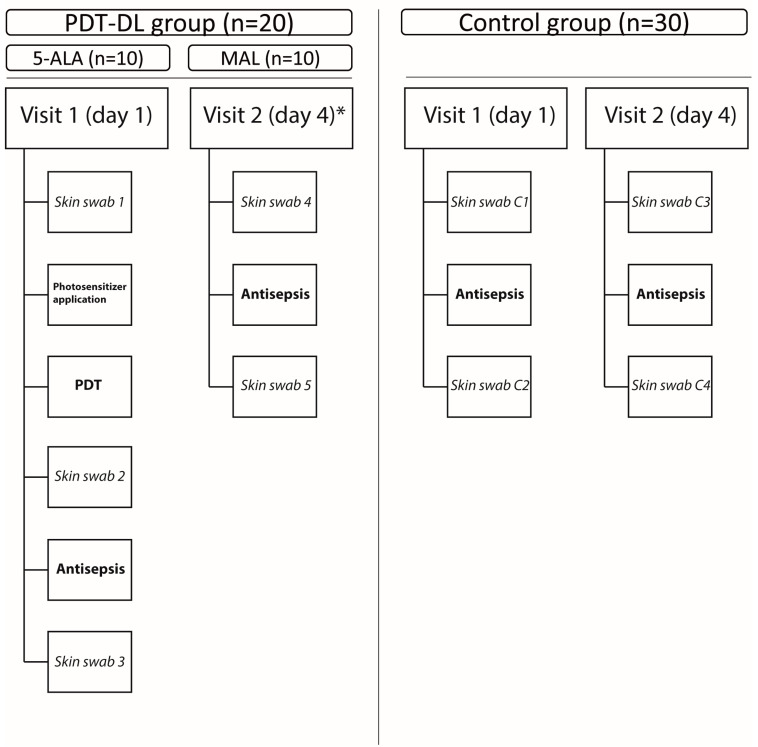
Overview of the study design. Interventions (bold), diagnostic procedures (italic) and timeframe according to group and treatment arm. * Only n = 20 participants were selected for these procedures on day 4. PDT-DL—photodynamic therapy with artificial daylight; 5-ALA—5-aminolevulinic acid; and MAL—methyl aminolevulinate.

**Figure 2 microorganisms-13-00204-f002:**
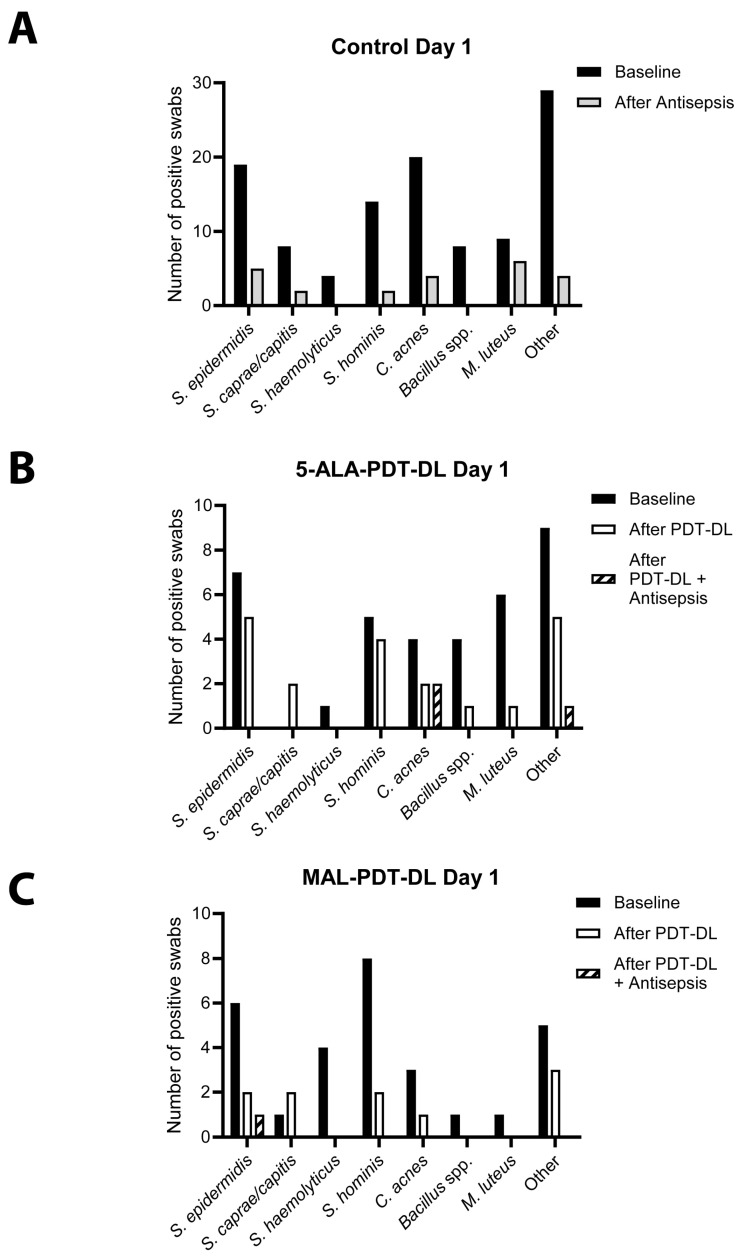
Effect of PDT-DL on specific bacterial species at day 1. (**A**) Number of positive samples on day 1 for the control arm at baseline and after antisepsis, as well as (**B**) the 5-ALA-PDT-DL arm and (**C**) the MAL-PDT-DL arm at baseline, after PDT-DL and after additional antisepsis. Other included *Corynebacterium* spp., *Staphylococcus warneri/pasteuri*, *Staphylococcus* spp., *Moraxella osloensis*, *Cutibacterium avidum*, *Staphylococcus cohnii*, *Kocuria* spp., *Dermabacter* spp., *Facklamia hominis*, *Staphylococcus lugdunensis*, *Pseudomonas* spp., Gram-positive rods, *Bacillus subtillis*, *Bacillus pumilus*, *Enterococcus faecium*, *Staphylococcus saprophyticus*, *Enterococcus faecalis*, *Bacillus megaterium*, *Staphylococcus. aureus*, *Cutibacterium* spp. and *Enterococcus gallinarum*. PDT-DL—photodynamic therapy with artificial daylight; 5-ALA—5-aminolevulinic acid; MAL—methyl aminolevulinate.

**Figure 3 microorganisms-13-00204-f003:**
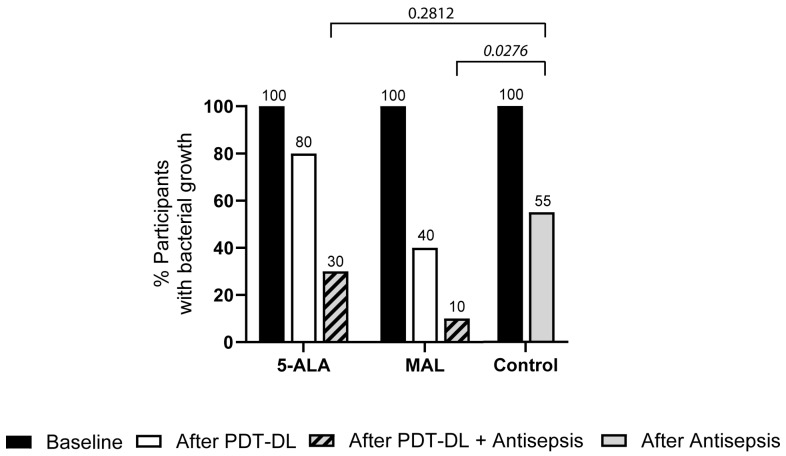
Overall effect of PDT-DL on bacterial growth. Percentage of participants with positive bacterial cultures directly after treatment. See Section 2 for further information. Numbers above the bars indicate % of participants with bacterial growth. Statistical significance was determined by two-sided Fisher’s exact test: the *p*-value for each comparison of 5-ALA—after PDT-DL + antisepsis and MAL—after PDT-DL + antisepsis against control—after antisepsis is indicated above the brackets. PDT-DL—photodynamic therapy with artificial daylight; 5-ALA—5-aminolevulinic acid; MAL—methyl aminolevulinate.

**Figure 4 microorganisms-13-00204-f004:**
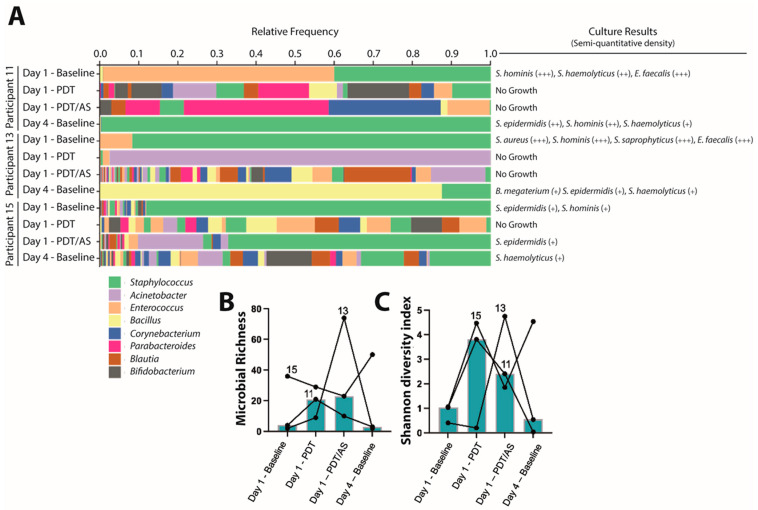
Skin bacteriome composition and alterations by MAL-PDT-DL. (**A**) The relative frequencies of the genus-level taxa are plotted. Each bar represents a participant ID and time point of sampling (i.e., day 1 baseline, day 1 after MAL-PDT-DL, day 1 after MAL-PDT-DL and additional antisepsis and day 4 baseline). The bars are grouped by participant ID from the earliest to latest time point. Only the top 8 most abundant genus-level taxa are shown in the legend. Culture results obtained from each sample are annotated to the right of the bar graph. +—sporadic (1000–10,000 CFUs); ++—moderate (10,000–100,000 CFUs); +++—abundant (100,000–1,000,000 CFUs). The (**B**) microbial richness (the number of unique genus-level taxa) and (**C**) Shannon diversity index (a combined measure of rich-ness and relative abundance) are plotted at each time point for each patient and connected by a line to demonstrate trends. The medians are plotted with solid bars. Each line is indicated with a patient ID. MAL-PDT-DL—photodynamic therapy with artificial daylight using MAL—methyl aminolevulinate; PDT—photodynamic therapy; and PDT/AS—photodynamic therapy followed by standard skin antisepsis.

**Table 1 microorganisms-13-00204-t001:** Adverse effects during and after 5-ALA-PDT-DL and MAL-PDT-DL in 10 participants each. Numbers in the brackets () mean % of participants.

Adverse Effect	Number of Participants (%)	
	5-ALA-PDT-DL	MAL-PDT-DL
Pain	0 (0)	0 (0)
Itching during PDT	2 (20)	0 (0)
Erythema after PDT	4 (40)	0 (0)
Itching + Erythema	1 (10)	0 (0)
No adverse effect	6 (60)	10 (100)

PDT—photodynamic therapy; 5-ALA-PDT-DL—photodynamic therapy with artificial daylight using 5-aminolevulinic acid; MAL-PDT-DL—photodynamic therapy with artificial daylight and methyl aminolevulinate.

## Data Availability

The original contributions presented in this study are included in the article/Appendix A. Further inquiries can be directed to the corresponding author.

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
