# Peer review of "Photodynamic Therapy with Protoporphyrin IX Precursors Using Artificial Daylight Improves Skin Antisepsis for Orthopedic Surgeries"

_microorganisms, 2025, doi:10.3390/microorganisms13010204_

Round 1

Reviewer 1 Report

Comments and Suggestions for Authors

The manuscript describes the effects of PDT using simulated daylight plus classical antisepsis on skin microorganisms. It is an interesting trial well designed and described, and the results obtained are highly relevant.

I have just one comment about the discussion of the results. It is mentioned that bacterial growth on the skin returns to basal levels four days after the different treatments. In line with this, it is stated that the effect of PDT-DL is not long-lasting, and that rapid recolonization occurs from deeper or surrounding skin. Is it expected to have a long-lasting bacterial decrease? It sounds inevitable that bacteria recolonize skin, so the necessity of repeated treatments may be considered in the discussion.

Minor issues:

- References 13 and 19 are cited without brackets in the text in lines 348,352, 356 and 371.

- The text in the axis of Figures 4A, B, and C can be increased for a better understanding.

Reviewer 2 Report

Comments and Suggestions for Authors

REVIEW OF THE ARTICLE BY TIZIANO A. SCHWEIZER ET AL.  ENTITLED "PHOTODYNAMIC THERAPY WITH PROTOPORPHYRIN IX PRECURSORS USING ARTIFICIAL DAYLIGHT IMPROVES SKIN ANTISEPSIS FOR ORTHOPAEDIC SURGERIES" (microorganisms-3358434 )

Schweizer et al. studied the effect of photodynamic therapy (PDT) mediated by 5-aminolevulinic acid and methyl-aminolevulinate on the skin bacteriome. Aminolevulinic acid and its derivatives are precursors of porphyrin molecules, which are powerful photosensitisers. These compounds are widely used in PDT for cancers and non-malignant skin diseases. The topic is relevant and within the scope of the journal. The authors investigated the diversity and total abundance of skin bacteria, focusing on Staphylococcus spp. and Cutibacterium spp., which are the most abundant genera on human skin. They employed both classical microbiological techniques and 16S rRNA metabarcoding. In the former approach, bacteria were identified using MALDI-MS. The data could be valuable for specialists in dermatology and microbiology. Although the topic is interesting and the data are important, the manuscript is poorly written, with serious flaws in all sections of the work. Therefore, it requires extensive editing before it can be reconsidered for publication. Specific points are provided below.

1. Abstract should be shortened. Recommended length  is no more than 250 words.

2. The text is poorly edited. Subsections are not numbered. Numbering of pages is incorrect. There is a lot of empty space on the pages. Font in Table 1 does not correspond to the journal’s template. Reference style does not correspond to the journal’s rules. Figures and tables should appear after their first mention in the text.

3. Abbreviations, presented in figures and tables, should be described in the legends (5-ALA-PDT-DL, MAL-PDT-DL, PDT, AS, PDT-DL, MAL, 5-ALA).

4. Number of references is extremely low for a work in microbiology. Minimum number of references is normally 40-60.

5. l. 50, 51, 172-174, 179, 181-182, 191, 212, 216, 287-293, 339, 367-368, 376. Species and genera names should be italicised.

6. l. 64, 72, 118. cm2 should be cm2.

7. Introduction is poor and does not give necessary background for the work. It is necessary to summarise an experience of the use ALA in PDA (see, e.g. Pignatelli et al. (2023) Int. J. Mol. Sc. 24(10), 8964 and other related works). Physico-chemical basis and its relationships with porphyrins also should be briefly described. The perspectives of the use of porphyrins for PDA should be elucidated (see König et al. (1992) Opt. Eng. 31, 1470–1475; Yeung et al. (2007) Lasers Surg. Med. 39, 1–6). Spectral characteristics of PpIX (absorbance/emission) should be discuss to justify its application in PDA (Chekanov et al. (2024) Life 14(10), 1271).

8. l 83-84. “Fitzpatrick skin phototype V & VI were also excluded” - why? Please, explain. Is high melanin content a limitation of the PDA?

9. Ethics: did the participants approve an information consent? Were the recommendations of the current version of the Declaration of Helsinki (World Medical Association. (2013). World Medical Association Declaration of Helsinki: ethical principles for medical research involving human subjects. JAMA 310(20), 2191-2194) taken into account?

10. Countries and cities should be indicated for ALL manufacturers.

11. A couple of times in the methods MALDI-TOF-MS is mentioned (l. 105, 131). The details of this method should be described in a separate subsection. If the analysis was done in a side organisation, it is not a reason not to describe it in the article. It is necessary to contact a service laboratory to clarify the details.

12. l. 131. The abbreviation should be introduced early (in l. 105).

13. l. 103. eSwab should be ESwab.

14. l. 104. “Swabs were cultured” sounds unscientific. Do you mean “Microorganisms from the swab were cultured”?

15. l. 116. Please, describe the “day light” (daylight??)  in more detail or provide emission spectra: the position of emission maxima is important

16. l. 122. “one-minute interval in between” - “one-minute interval”?

17. l. 135. Should be added to the reference list.

18. l. 104. The abbreviation CFU should be introduced here.

19. l. 140. PCR profile should be described.

20. l. 148. Should be described above (l. 140).

21. l. 143. Manufacturer? Country? City?

22. l. 150-160. References should be provided for the software: QIIME2 v. 2024.2 (Bolyen et al. (2019) Nat. Biotechnol. 37:852–857), DADA2 (Callahan et al. (2016) Nat. Methods 13(7), 581-583), SILVA v 138 (Quast et al. (2013) Nucl. Acids Res. 41 (D1): D590-D596).

23. l. 153. Please, indicate the release of SILVA (not version only).

24. l. 152. bayes should be Bayes.

25. l. 150-160. Should be described in more detail: how trimming was performed, how the “technical” sequences were eliminated. How can you explain the presence of chloroplast reads in the skin datasets (l.155)? Were mitochondrial reads and other eukaryotic reads eliminated (Chekanov et al. (2019). J. Euk. Microbiol. 66(5), 853-856)?

26. l. 160. … as histograms?

27. Please, describe clearly the statistical tests used. For example, why did you use the exact Fisher test, which is commonly used for small datasets as an analogue of the χ2-test. Was the normality tested? Why was not the ANOVA used for the multivariate datasets?

28. I cannot understand the strategy of statistical treatment from the figures. Why only two groups of columns are united (Figure 2A). What do the numbers above brackets mean? What does it mean? Why are there not any treatments in Figure 2B, Figure 3, and Figure 4(B,C)? The χ2-test can be used to describe the difference between the control and treatment (Figure 3). Yates's correction for continuity can be used for datasets with n<10.

29. l. 171. skin of the participants?

30. l. 190. “were heterogeneous” - what does it mean? “After PDT-DL and/or antisepsis, multispecies bacterial communities were found on the skin”?

31. l. 176-192. Which results are presented here, 16S rRNA metabarcoding or MALDI mass?

32. Table 1: what do the numbers in the brackets mean?

33. Through the text: do you mean “bacteriome” instead of “microbiome”, because you focused on bacteria only?

34. l. 261. % should be the percentages. % from what? two-tailed Fisher?

35. l. 284-287. In my version of the manuscript, I cannot distinguish any colours except black and white.

36. l. 313. What do you mean by richness?

37. l. 29, 58, 185, 187, 258, 324, 325, 328, 394. “showed positive cultures” - what is it?

38. Figure 4A: where is Cutibacterium?

39. l. 352, 356. A typing error?

40. Skin bacteria, such as C. acnes and S. epidermidis, can produce porphyrins. It is necessary to discuss their possible contribution to the photodamage in the process of PDA (Chekanov et al. (2024) Life, 14(10), 1271).

41. When discussing bacterial diversity, it is also important to discuss a possible competition of different species, i.e. destroying one bacteria, such as Staphylococcus, can lead to a massive development of others, such as Cutibacterium (see and cite Claudel et al. (2019) Dermatol, 235, 287–294).

42. l. 372-384.You should explain the absence of Cutibacterium spp.in the metabarcoding data.

43. l. 385-390. It is also important to note a possible  incompatibility of current taxonomy and information in the metagenomic databases.

44. l. 385. “Despite” should be “despite”.

45. l. 385-406. Reference(s) required.

Comments on the Quality of English Language

I understand the text, but it should be extensively editted at the stage of English correction. I tried to point out some issues (see Comments and Suggestions for Authors), but there are many other concerns, e.g. articles using. 

Round 2

Reviewer 2 Report

Comments and Suggestions for Authors

I have carefully reviewed the revised manuscript and your responses to my comments. Thank you for the detailed explanations and the corrections made to the text. The article has been significantly improved. However, some issues remain unresolved or have been insufficiently addressed. In particular, although you provide detailed explanations in the cover letter, there are no corresponding corrections in the manuscript. Below, I provide explanations of my original comments (referenced by their numbers from the first review round), which I have to reiterate.

3. Abbreviations, presented in figures and tables, should be described in the legends (5-ALA-PDT-DL, MAL-PDT-DL, PDT, AS, PDT-DL, MAL, 5-ALA) - it has not been done in Figure 1.

8. I understand your answer, but it has not been commented in the text.

11. I am afraid you did not understand my comment regarding MALDI-ToF-MS. You provide references to the theoretical basis of the method (references [34, 35]), which is textbook knowledge, whereas I was asking about the experimental details of your work. MALDI-MS is a complex instrumental method in which the results strongly depend on the parameters used. These parameters must be indicated, especially the detection mode (cation/anion), m/z detection range, laser type and frequency, software used for the analysis, scoring criteria (Log(score) thresholds for species-level and genus-level identification), and the reference database employed for identification. The last two points are particularly important for the results of identification. Please see examples of MALDI-MS descriptions in similar papers, such as 10.1128/JCM.00626-12 and 10.1016/j.anaerobe.2022.102554. If the analysis was conducted by a third-party organisation, this is not a valid reason to omit its description in the article. It is necessary to contact the service laboratory to clarify the details.

19. I am afraid you did not fully understand my comments regarding the PCR profile. The PCR profile refers to its specific conditions, including the model of your amplifier, the temperatures and durations of initial denaturation, cyclic denaturation, annealing, elongation, final elongation, and the number of cycles. The primers used should also be specified. If the analysis was conducted by a third-party organisation, this is not a valid reason to omit its description in the article. It is necessary to contact the service laboratory to clarify the details.

25. I am afraid you did not fully understand my comments regarding the appearance of chloroplast reads. I did not ask you to compare them with other studies but to explain their presence in your dataset. Please note that 'chloroplast' should not be capitalised.

27-28. I understand your detailed answers, but there are no corresponding explanations in the text, especially 'Our interest lies in the difference between the above mentioned comparison, i.e. the final treatment. Anything else appears to be not relevant for any clinical application of the proposed treatment'. How was normality assessed? What was an output of this test?

32. Please, indicate clearly: 'the numbers in the bracket mean % of participants'.

38, 42. I understand your detailed answer, but it should be explained in thetext.

44. l. 444. “Despite” should be “despite”.

Author Response

Reply to Reviewer 2

I have carefully reviewed the revised manuscript and your responses to my comments. Thank you for the detailed explanations and the corrections made to the text. The article has been significantly improved. However, some issues remain unresolved or have been insufficiently addressed. In particular, although you provide detailed explanations in the cover letter, there are no corresponding corrections in the manuscript. Below, I provide explanations of my original comments (referenced by their numbers from the first review round), which I have to reiterate.

  1. Abbreviations, presented in figures and tables, should be described in the legends (5-ALA-PDT-DL, MAL-PDT-DL, PDT, AS, PDT-DL, MAL, 5-ALA) - it has not been done in Figure 1.

We thank the reviewer for pointing out this flaw and added abbreviations to all figures.

  1. I understand your answer, but it has not been commented in the text.

As required by the reviewer, we added the following statement to Lines 110-112: The exclusion of patients with skin phototypes V and VI is due to potentially higher risk of developing inflammatory reactions to light exposure as well as the potentially thicker skin layers, potentially limiting the effect of PDT to reach deep-seated skin bacteria. Since this is based on observations at the Department of Dermatology during routine treatment of acne, no references can be cited.

  1. I am afraid you did not understand my comment regarding MALDI-ToF-MS. You provide references to the theoretical basis of the method (references [34, 35]), which is textbook knowledge, whereas I was asking about the experimental details of your work. MALDI-MS is a complex instrumental method in which the results strongly depend on the parameters used. These parameters must be indicated, especially the detection mode (cation/anion), m/z detection range, laser type and frequency, software used for the analysis, scoring criteria (Log(score) thresholds for species-level and genus-level identification), and the reference database employed for identification. The last two points are particularly important for the results of identification. Please see examples of MALDI-MS descriptions in similar papers, such as 10.1128/JCM.00626-12 and 10.1016/j.anaerobe.2022.102554. If the analysis was conducted by a third-party organisation, this is not a valid reason to omit its description in the article. It is necessary to contact the service laboratory to clarify the details.

As requested by the reviewer, we added the details about our applied MALDI-ToF-MS identification. We have therefore restructured subchapters 2.2 Skin swabs and 2.5 Microbiological analysis in order to describe the identification process from culture to mass spectrometry more logically. We deleted the MALDI information from subchapter 2.2 and added the detailed information to subchapter 2.5.

  1. I am afraid you did not fully understand my comments regarding the PCR profile. The PCR profile refers to its specific conditions, including the model of your amplifier, the temperatures and durations of initial denaturation, cyclic denaturation, annealing, elongation, final elongation, and the number of cycles. The primers used should also be specified. If the analysis was conducted by a third-party organisation, this is not a valid reason to omit its description in the article. It is necessary to contact the service laboratory to clarify the details.

As requested by the reviewer, we added the PCR details in subchapter 2.6 Metagenomic analysis. Unfortunately, we cannot publish the primer sequences used for sequencing as these are proprietary. The applied primers belong to the QIAseq 16S/ITS kit and are not published by Qiagen.

  1. I am afraid you did not fully understand my comments regarding the appearance of chloroplast reads. I did not ask you to compare them with other studies but to explain their presence in your dataset. Please note that 'chloroplast' should not be capitalised.

We apologize for the misunderstanding. In our specific dataset, chloroplast sequences were present in just two samples. One sample timepoint from patient 13 (83 sequences total) and one sample from timepoint patient 15 (27 sequences total). These sequences were present in low abundance representing approximately 0.1% of all sequences from each sample respectively.  One explanation is that the chloroplast sequences are false positives due to their low abundance and the fact that they are only present in just two samples across the dataset. Another explanation is that the chloroplasts sequences are truly present in the two samples as they were not detected in the negative control and thus likely not a contaminant from sample processing. As our study was designed to assess the bacterial microbiome it is difficult to explain further the presence of chloroplast sequences in these two samples.

We have removed the comparison with the study by Kramer et al. and describe that the occurrence of chloroplasts is likely false-positive. Line 216-220: “Amplicon sequence variants (ASVs) classified as chloroplasts were removed as these are likely false positives due to their presence in abundances of approximately 0.1% in two patient samples. Low frequency or rare ASVs (those that contribute less than 0.1% of the average sequencing depth) were also removed.”

27-28. I understand your detailed answers, but there are no corresponding explanations in the text, especially 'Our interest lies in the difference between the above mentioned comparison, i.e. the final treatment. Anything else appears to be not relevant for any clinical application of the proposed treatment'. How was normality assessed? What was an output of this test?

We thank the reviewer for opening this dialogue. We do not choose a single normality test but drew a conclusion from four different tests, each with its strengths and weaknesses. This approach is necessary under certain conditions, normality tests can fail to provide the correct results. As stated earlier in the manuscript, these tests were performed using GraphPad Prism, and the outputs from these tests answer the question of normal distribution, i.e., whether the data are normally distributed—yes or no.

We have added this section to Lines 227-230: “Data was assessed for normality (Gaussian distribution) based on a conclusion drawn from results obtained from a Shapiro-Wilk, Anderson-Darling, D’Agostino-Pearson and Kolmogorov-Smirnov normality test.”

  1. Please, indicate clearly: 'the numbers in the bracket mean % of participants'.

We added this sentence to Line 392: “Numbers in the brackets () mean % of participants.”

38, 42. I understand your detailed answer, but it should be explained in thetext.

We added this sentences to Lines 292-2294, which addresses both of the points raised: “Importantly, while Cutibacterium spp. was detected, it was so at extremely low abundance and therefore not represented. Also, culture of these samples were nega-tive for growth of Cutibacterium spp.”

  1. l. 444. “Despite” should be “despite”.

Was adjusted in Line 489.